# Modulation between capacitor and conductor for a redox-active 2D bis(terpyridine)cobalt(II) nanosheet via anion-exchange
Kenji Takada [1] ✉, Miyu Ito [2], Naoya Fukui [1] & Hiroshi Nishihara [1,2] ✉

Ionic polymers are intriguing materials whose functionality arises from the synergy between ionic polymer backbones and counterions. A key method for enhancing their functionality is the post-synthetic ion-exchange reaction, which is instrumental in improving the chemical and physical properties of polymer backbones and introducing of the functionalities of the counterions. Electronic interaction between host polymer backbone and guest ions plays pivotal roles in property modulation. The current study highlights the modulation of responses to external electric field in cationic bis(terpyridine)cobalt(II) polymer nanofilms through anion-exchange reactions. Initially, as-prepared chloride-containing polymers exhibited supercapacitor behaviour. Introducing anionic metalladithiolenes into the polymers altered the behaviour to either conductive or insulative, depending on the valence of the metalladithiolenes. This modulation was accomplished by fine tuning of charge-transfer interactions between the bis(terpyridine)cobalt(II) complex moieties and redox-active anions. Our findings open up new avenue for ionic polymers, showcasing their potential as versatile platform in materials science.

Recent advances in molecular-based thin-film materials have sparked considerable interests owing to their structural and functional versatility and applicability for conventional materials science[1–12]. These thin films, including both covalent bond-based (such as polymer or covalent organic framework thin films) and coordination bond-based (such as metal-organic frameworks thin films or coordination nanosheets) thin films, have been enthusiastically investigated across various materials science field. The diversity in component combinations significantly contributes to the high functionality of these molecular-based polymer thin films. In ionic polymer-based thin films, charge-compensating counterions in the thin films can enhance their functionality[13–20].

Ion-exchange reactions are recognized as a straightforward method for post-synthetic functionalization of ionic polymers. Replacing counterions in the polymer backbone with different ions holds promise not only for introducing functionalities of counterions but also for altering the chemical and physical properties through polymer-ions interactions. Many studies have focused on enhancing the functionalities such as luminescence, catalytic activity, and electronic conductivity, in response to stimuli and incorporating the counterion functionalities into thin films[21–30]. However, a more intrinsic control over functionality, fundamentally altering the response of an ionic polymer film to the same stimulus, has yet to be achieved. Thus, developing multi-functional ionic polymers with precise functional tunability via ion-exchange is crucial.

Herein, we detail the modulation of responses to external electric field for a redox-active bis(terpyridine)cobalt(II) polymer (**1**) via anion-exchange strategy (Fig. 1). This polymer exhibits a redox-state-dependent physical properties such as electrochromism and redox conduction[31,32]. We demonstrate the efficient replacement of chloride ions in **1** with redox-active bis(maleonitriledithiolato)nickelate anions ([Ni(mnt)$_2$]$^{n-}$, $n = 1$ or 2), and investigate their responses to external electric field in solid state on interdigitated array (IDA) electrodes. Cl$^-$, [Ni(mnt)$_2$]$^-$, and [Ni(mnt)$_2$]$^{2-}$ anions differ in size and redox potentials, so that differences in the diffusion of the anions in the cationic polymer framework or electronic charge transfer interaction between the cationic polymer framework and the anions results in the modulation between capacitor and conductor. Our findings suggest that anion-exchange reaction with bis(terpyridine)cobalt(II) polymers is a practical method for precise ex-situ control of their electronic functions.

[1]Research Institute for Science and Technology, Tokyo University of Science, 2641, Yamazaki, Noda, Chiba, 278-8510, Japan. [2]Faculty of Science and Technology, Tokyo University of Science, 2641, Yamazaki, Noda, Chiba, 278-8510, Japan. ✉e-mail: takada.k.ag@rs.tus.ac.jp; nisihara@rs.tus.ac.jp

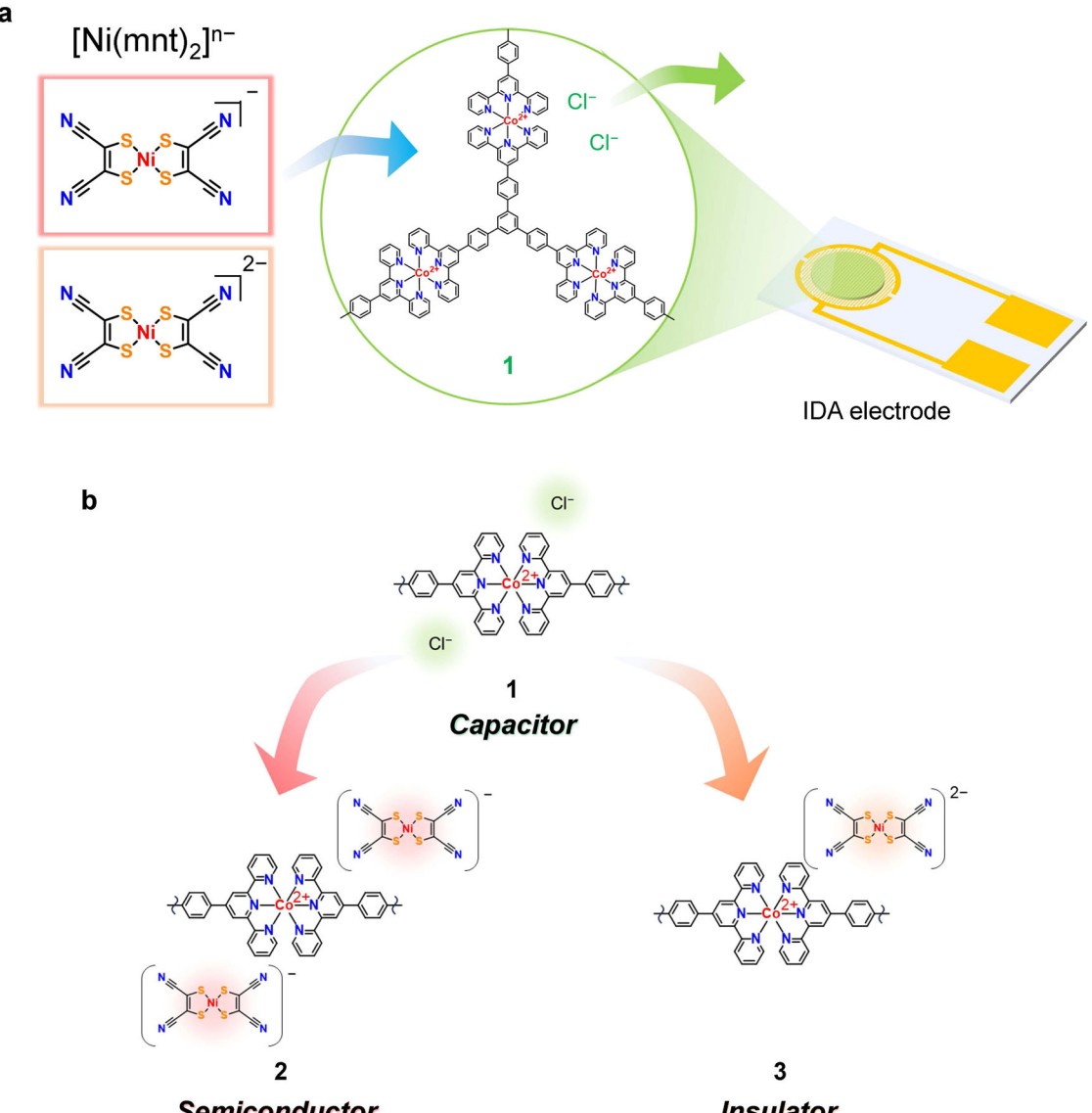

**Fig. 1 | Anion-exchange-induced modulation of responses to external electric field in 1. a** Schematic illustration of anion-exchange reaction of **1** conducted on IDA electrode. **b** Modulation between capacitor and conductor by introduction of the redox-active [Ni(mnt)₂]ⁿ⁻ via anion-exchange reaction.

## Results

### Anion-exchange of 1

We investigated anion-exchange characteristics of **1**, prepared using the liquid-liquid interfacial coordination reaction technique established for the synthesis of coordination polymer nanofilms[31–36]. **1** was obtained as an orange film, formed at the liquid-liquid interface. Energy dispersive X-ray spectroscopy under scanning electron microscope observation (SEM/EDS) revealed that **1** includes uniformly distributed constituting elements, C, N, Co, and Cl (Supplementary Fig. 1a). Raman spectrum showed the C≡N stretching peak at 1585 cm⁻¹, shifted from that of the free terpyridine ligand by 18 cm⁻¹ (Supplementary Fig. 1b). X-ray photoelectron spectroscopy (XPS) gave the same spectra for **1** as the previous study (Supplementary Fig. 1c)[31]. The cyclic voltammograms of **1** (Supplementary Fig. 2) showed intense reversible redox waves at −1.13 V vs. (ferrocenium/ferrocene) Fc⁺/Fc and −1.88 V vs. Fc⁺/Fc, attributed to the [Co(tpy)₂]²⁺/[Co(tpy)₂]⁺ and [Co(tpy)₂]⁺/[Co(tpy)₂]⁰ redox couples, respectively. The voltammogram also showed weak reversible redox wave at −0.13 V vs. Fc⁺/Fc derived from the [Co(tpy)₂]³⁺/[Co(tpy)₂]²⁺ redox couple. These characterization data were identical to the previous reports[31], and confirmed the preparation of **1**.

Anion-exchange to [Ni(mnt)₂]ⁿ⁻ was completed by simply immersing a **1** film into (nBu₄N)ₙ[Ni(mnt)₂] solutions (n = 1, 2), which afforded anion-exchanged polymers containing [Ni(mnt)₂]⁻ (**2**) and [Ni(mnt)₂]²⁻ (**3**) (Fig. 2a, Supplementary Table 1). The optimal solvents identified for the anion-exchange process were CH₃CN and C₂H₅OH for (nBu₄N)₂[Ni(mnt)₂] and (nBu₄N)[Ni(mnt)₂], respectively. After the reaction, the polymer films exhibited coloration due to the incorporated metalladithiolate ions (Fig. 2b). SEM/EDS confirmed the absence of Cl and presence of S and Ni in the polymer films for both monovalent and divalent anions (Fig. 2c). In SEM/EDS elemental mapping, uniform distribution of all constitute elements was evident (Fig. 2d,e). The S, Co, and Ni peak area analysis from the SEM/EDS indicated incorporation ratio of [Ni(mnt)₂]ⁿ⁻ ions in **2** and **3** of 0.55 : 1, aligning with the expected stoichiometric ratio (0.5 : 1) within the bounds of SEM/EDS semi-quantitative analysis accuracy. These findings affirm the efficacy of the anion-exchange reaction in **1**. The cross-sectional elemental mapping by scanning transmission electron microscopy confirmed that Cl was not detected in the polymer films (Supplementary Fig. 3), meaning that the anion-exchange reaction efficiently proceeded without the limitation at the surface of **1**. XPS further validated the anion-exchange of **1**,

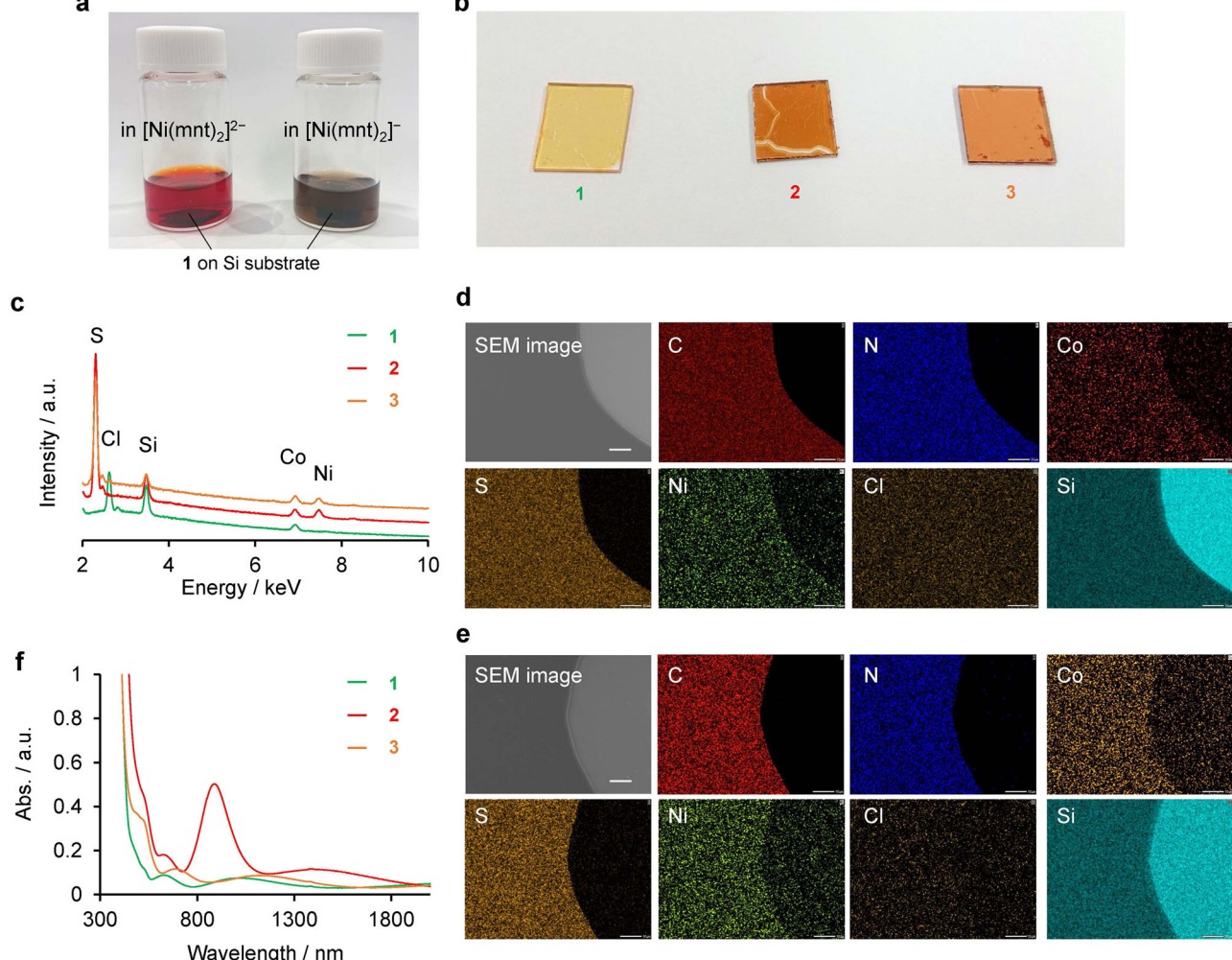

**Fig. 2 | Anion-exchange of 1. a** Photograph of anion-exchange reaction of **1**. **b** Photograph of **1**−**3**. **c** SEM/EDS spectra of **1**−**3**. **d**, **e** SEM images and SEM/EDS mappings of **2** and **3** (Scale bar: 10 μm). **f** UV-vis-NIR spectra of **1**−**3**.

evidenced by the disappearance of Cl 2p peak as shown in Supplementary Fig. 4. Instead, following the anion-exchange, peaks corresponding to S, and Ni appeared. The N 1s peak from tetrabutylammonium ions around 402 eV (Supplementary Fig. 4) was absent in the XPS, indicating no adsorption of the ion pairs onto the polymers. UV-vis-NIR spectroscopy (Fig. 2f) corroborated the predominant incorporation of the respective metalladithiolates into **2** and **3**. Specifically, the distinct peak at 889 nm, linked to the π-π* transition of the $[Ni(mnt)_2]^-$ ion, was observed in the spectrum of **2** (Supplementary Fig. 5). Conversely, **3** exhibited no NIR absorption band because the $[Ni(mnt)_2]^{2-}$ ion has no absorption in the NIR region. Atomic force microscopy (AFM) demonstrated that the polymer sheets thickened after the anion-exchange, suggesting the substitution of chloride ions with the bulkier metalladithiolene anions (Supplementary Fig. 6). The electrochemical rest potentials of **2** and **3** were −0.18 V and −0.30 V vs. $Fc^+/Fc$, respectively. Given that the redox potential of $[Ni(mnt)_2]^-/[Ni(mnt)_2]^{2-}$ was −0.27 V vs. $Fc^+/Fc$[37], the measured rest potentials are indicative of the oxidation states of the metalladithiolate ions involved in the anion-exchange reaction. These findings confirm the efficiency of anion-exchange process in **1**, whose electrochemical rest potential was −0.46 V vs. $Fc^+/Fc$[32].

Raman spectroscopy provided the electronic structure information about the polymer films. Raman spectra of **1**−**3** in the Raman shift below 2000 cm⁻¹ displayed similar peak patterns (Fig. 3a), indicating that the cationic polymer framework remained intact after the anion-exchange. In

the spectra after the anion-exchange, the C≡N stretching peaks of $[Ni(mnt)_2]^{n-}$ complexes observed at approximately 2200 cm⁻¹ were dependent on their oxidation states. Two peaks appeared at 2221 and 2200 cm⁻¹ for **2**, whereas a peak was observed at 2191 cm⁻¹ for **3** (Fig. 3b). According to the Raman spectra of $(nBu_4N)^+$ salts of $[Ni(mnt)_2]^{n-}$ complexes (Supplementary Fig. 7), C≡N stretching peak at 2215 cm⁻¹ was attributed to the monovalent anion, while the peak at 2191 cm⁻¹ was the divalent anion. Therefore, while the dithiolene complex exists as $[Ni(mnt)_2]^{2-}$ in **3**, both $[Ni(mnt)_2]^-$ and $[Ni(mnt)_2]^{2-}$ oxidation states coexist in the film in **2**. These findings imply a partial charge transfer between the $[Ni(mnt)_2]^-$ and $[Co(tpy)_2]^{2+}$ moieties.

This partial charge transfer is confirmed by XPS. In the XP spectrum of **2** in Fig. 3c, the S 2p peak was observed at 161.5 eV, which was slightly lower than those of $(nBu_4N)_2[Ni(mnt)_2]$ that appeared at 163.0 eV. This result indicates partial reduction of the monovalent nickelladithiolene complexes in the cationic polymer framework. Additionally, the Co 2p₃/₂ peak shifted to higher binding energy, meaning the partial oxidation of the $[Co(tpy)_2]^{2+}$ moieties. Therefore, electron transfer from $[Co(tpy)_2]^{2+}$ to $[Ni(mnt)_2]^-$ occurred after the anion-exchange. These peak shifts on XP spectra were not observed in **3**, suggesting no charge transfer interaction between $[Co(tpy)_2]^{2+}$ to $[Ni(mnt)_2]^{2-}$ (Fig. 3c).

To investigate the driving force of the anion-exchange reaction from chloride to $[Ni(mnt)_2]^{n-}$, the inverse anion-exchange reaction was performed for **2** and **3** with 5 mM $nBu_4NCl$ solution. The SEM/EDS revealed that the coexistence of metalladithiolene anions and chloride ions after the

**Fig. 3 | Evaluation of the electronic states of 1–3.**
**a** Raman spectra of **1–3**. **b** Expansion of the Raman spectra in panel **a** featuring C≡N stretching peaks. **c** XP spectra of **2** and ($n$Bu$_4$N)[Ni(mnt)$_2$] in S 2p core level (left), and **1** and **2** in Co 2p core level (right).

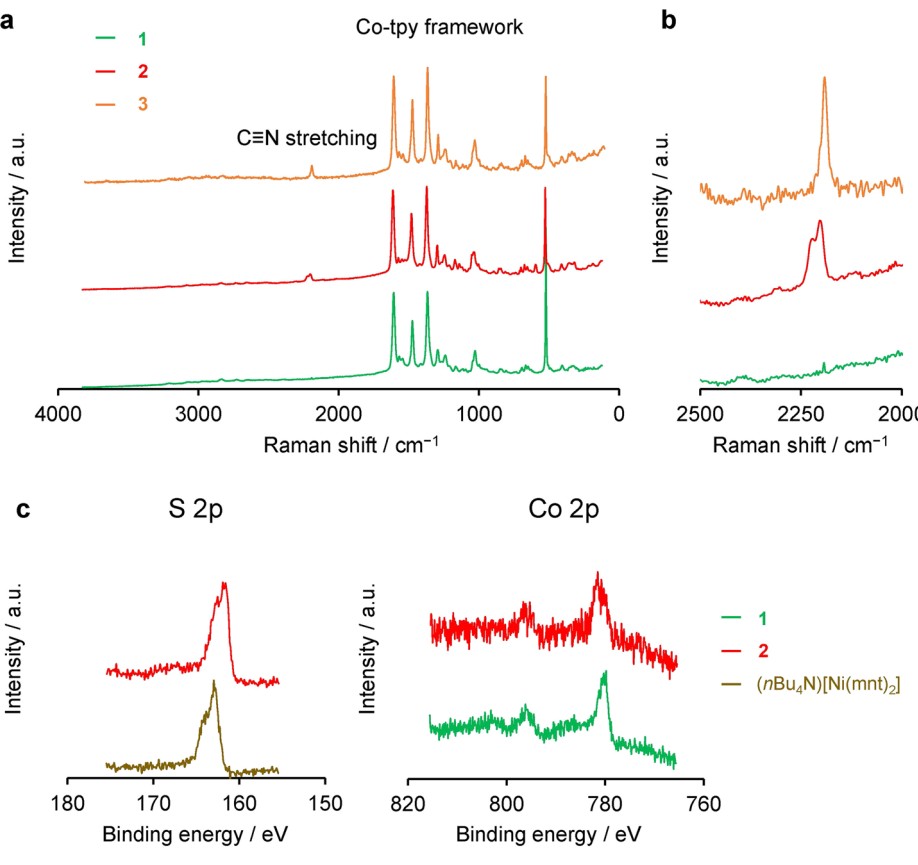

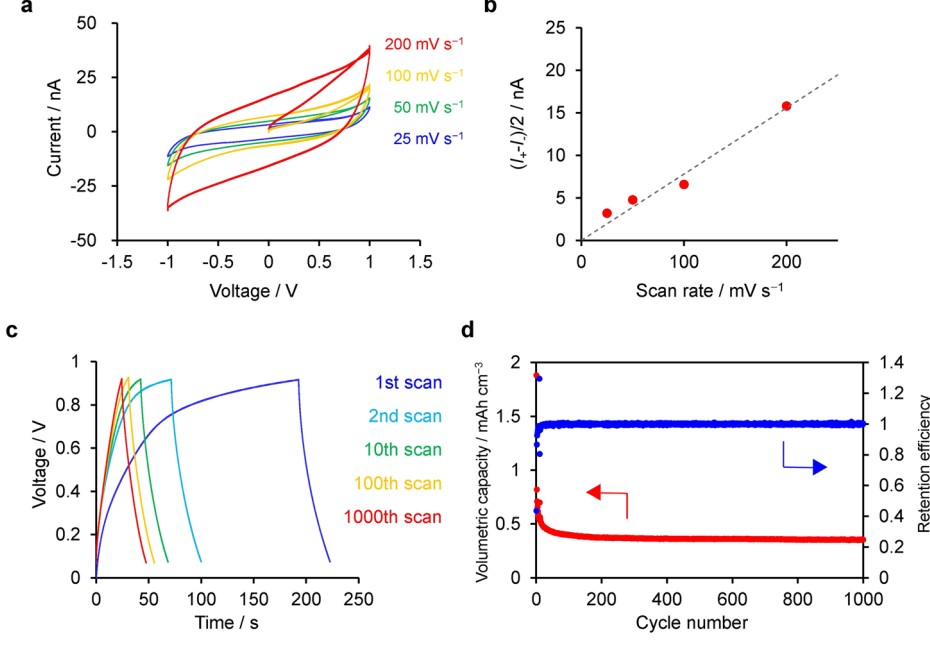

**Fig. 4 | Capacitive response of 1. a** *I-V* curves for **1** with different scan rate. **b** Scan rate-dependence of the width of *I-V* curves in **a** at 0 V. **c** Galvanostatic charge-discharge curves of **1** from 0 to +1 V recorded with 1 nA s$^{-1}$. **d** Cycle-dependence of volumetric capacity and retention efficiency.

reaction, with the approximately 60% and 70% anion-exchange, respectively (Supplementary Fig. 8). The excess Cl$^-$ did not replace the metalladithiolenes completely. These results indicated that the anion-exchange from chloride to [Ni(mnt)$_2$]$^{n-}$ was a thermodynamically favoured reaction. π-π interactions and charge-transfer interactions between the bis(terpyridine) cobalt(II) polymer backbone and the metalladithiolenes are preferable while

chloride-π interactions were less interactive[38], which is the plausible origin of the driving force to the anion-exchange reaction.

## Capacitive response of 1

The response of **1** to external electric field was examined using IDA electrodes in a dry condition without additional supporting electrolyte

**Fig. 5 | Electrical conductivity of 2 and 3. a** *I-V*
curves for **2** and **3**. **b** Arrhenius plot detailing the
temperature-dependent conductivity of **2**.
**c** Schematic illustration of the plausible conductivity
mechanism based on electron hopping between
partial charge transfer metal complex sites in **2**.
Electron hopping between redox active $[Co(tpy)_2]^{m+}$
(m = 2 or 3) and $[Ni(mnt)_2]^{n-}$ sites is responsible for
electron transport. The colours of each metal com-
plex site depict the difference in the oxidation states.

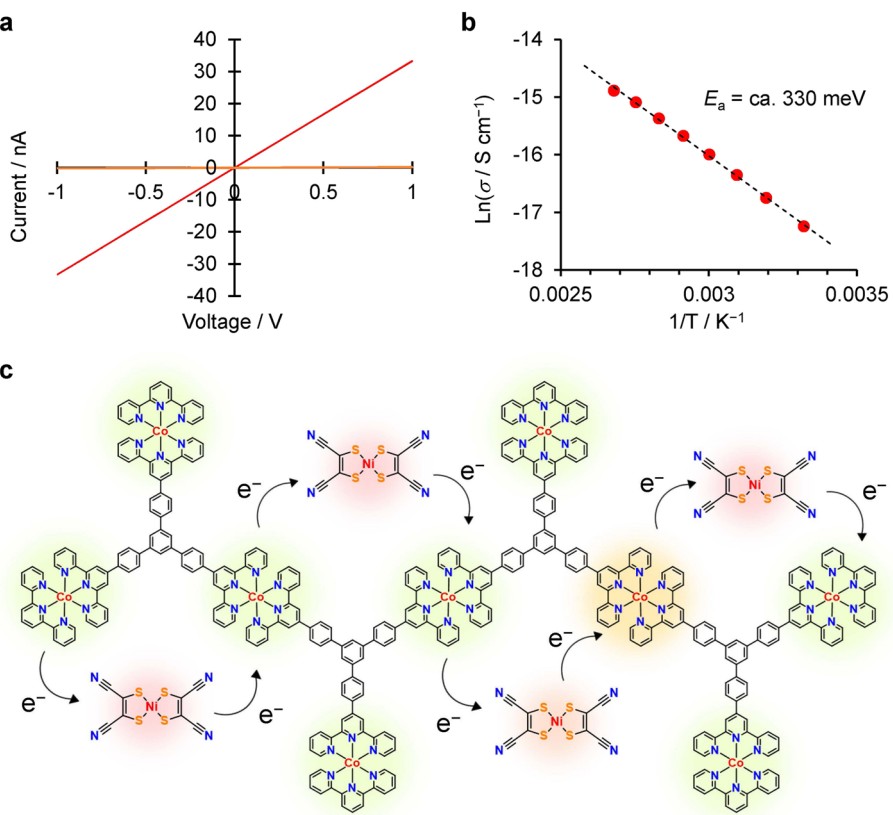

(Supplementary Fig. 9). Figure 4a shows the representative *I-V* curves of **1**
between −1 and +1 V with varying scan rates. The *I-V* curves are dependent
on the scan rate. The width of the *I-V* curves increases proportionally to the
scan rate (Fig. 4b). This relationship is a characteristic of the charge/dis-
charge dynamics typical for an electrochemical supercapacitor and thought
to stem from movement or displacement of chloride ions within the poly-
mer framework when external electric field is applied. Based on the width of
electrochemical double layers, the areal and volumetric capacitance of the **1**
was estimated as $7.8 \pm 4.0\ \mu F/cm^2$ and $0.19 \pm 0.09\ F/cm^3$, respectively.
Electrochemical impedance spectroscopy (EIS) further elucidated the
capacitive nature of **1**. The resulting spectrum was modelled using an
equivalent circuit that included contact and film resistance (R), charge
transfer resistance ($R_1$), Warburg resistance for the diffusion of $Cl^-$ ($W_{diff}$),
capacitance of the **1** film ($C_1$), leak resistance ($R_2$), and pseudocapacitance
for the faradaic process ($C_2$) as shown in Supplementary Fig. 10[39,40]. The
low-frequency phase angle approached approximately 45°, indicative of
diffusion-controlled process. The capacitance $C_1$ was consistent to that
measured from the cyclic voltammetry. Galvanostatic charge/discharge
cycles revealed rapid decrease of capacity within the initial cycles, stabilizing
at ca. 25% of the initial capacity (Fig. 4c, d). While these capacity and stability
metrics are modest, our findings confirm the potential of chloride-
containing bis(terpyridine)metal(II) polymers as solid-state electrolyte for
all-solid-state supercapacitor[41,42]. This paves the way for the future
enhancements in microsupercapacitors based on $M(tpy)_2$ complexes
through structural refinement.

**Conductive response of 2 and 3**
The response to the external electric field after the anion-exchange was
investigated using IDA electrodes. Anion-exchange reaction was performed
for film **1** immobilized on IDA electrodes, and the conductivity measure-
ments were performed by the two probe method at room temperature (ca.
300 K). Figure 5a depicts the *I-V* curves −1 V and +1 V for **2**. While the
initial **1** responded as a microsupercapacitor, **2** exhibited the almost linear *I-*

*V* curve with the conductivity of $1.1 \pm 0.2 \times 10^{-8}\ S\ cm^{-1}$. Temperature-
dependent conductivity measurement revealed that the conductivity of **2**
decreased with increasing temperature, suggesting the semiconductive
nature of **2** (Fig. 5b). From the Arrhenius plot in Fig. 5b, the activation
energy ($E_a$) was calculated to be $0.33 \pm 0.01\ eV$. These results indicated that
the anion-exchange from $Cl^-$ to $[Ni(mnt)_2]^-$ endowed the drastic change in
responses to external electric field.

Conversely, the conductivity of **3** falls below the detection limit of the
measuring apparatus ($<10^{-11}\ S\ cm^{-1}$), clarifying that **3** is an insulator, whose
conductivity is lower than that of **2** by 4 orders of magnitude. Additionally, the
*I-V* curves of **3** showed no dependence on the scan rates. Therefore,
$[Ni(mnt)_2]^{2-}$ anions neither function as electron conductors nor as electrolytes.
The capacitive difference between chloride and the divalent nickeladithiolene
complexes can stem from the relatively large volumetric size of $[Ni(mnt)_2]^{2-}$.

The modulation of responses to external electric field upon anion-
exchange can be explained by electronic interactions between the cationic
polymer backbone and anions. The conductive nature of **2** originates from
the charge-transfer between $[Co(tpy)_2]^{2+}$ and $[Ni(mnt)_2]^-$ moieties.
Because the redox potentials of $[Co(tpy)_2]^{3+/2+}$ (−0.15 V vs $Fc^+/Fc$) and
$[Ni(mnt)]^{-/2-}$ (−0.27 V vs $Fc^+/Fc$) redox couples are close to each other
($\Delta E = 0.12\ V$)[31,37], partial charge-transfer interactions from $[Co(tpy)_2]^{2+}$ to
$[Ni(mnt)_2]^-$ is expected, causing the mixed-valence states and thus inducing
electric conductivity to the **2** film through the redox-conduction mechanism
(electron hopping between redox active sites)[43,44]. Conversely, such charge-
transfer does not occur in **3**, resulting in its insulating behaviour.

The hypothesis on the partial charge-transfer model is also supported
by the response to external electric field of a bis(terpyridine)cobalt(II)
polymer impregnated with another monoanionic metalladithiolene com-
plex, $[Ni(tdt)_2]^-$ (tdt: 4-toluene-1,2-dithiolato). The $[Ni(tdt)_2]^-$-containing
polymer (**4**) was successfully prepared via anion-exchange reaction (Sup-
plementary Table 1 and Supplementary Figs. 11–13). The electric con-
ductivity of **4** was measured via the same procedure using IDA electrodes,
indicating that **4** was insulating with lower electrical conductivity than the

https://doi.org/10.1038/s42004-024-01274-4 **Article**

detection limit (Supplementary Fig. 14). The redox potential of $[Ni(tdt)_2]^-$/$[Ni(tdt)_2]^{2-}$ is $-0.95$ V vs. $Fc^+/Fc$[45,46], which does not match the oxidation potential of the polymer backbone ($-0.14$ V vs. $Fc^+/Fc$). The difference between the redox potentials of the cationic framework and $[Ni(tdt)_2]^-$ was ca. 0.8 V, indicating that the charge-transfer interaction was not expected. Therefore, the conductive behaviour stems from the host-guest charge-transfer interaction.

To investigate the conductivity mechanism of **2**, the I-V curve was measured with wider voltage range between $-6$ V and $+6$ V (Supplementary Fig. 15a). In the wider range, the current was not linear to the applied voltage (Supplementary Fig. 15b). This non-linear I-V curves were fitted with the simulation based on the redox conduction mechanism[47]. In addition, the potential-dependent conductivity measurement of **2** shows the conductivity increasing around $-0.1$ V vs. $Fc^+/Fc$, near to the $[Co(tpy)_2]^{3+}$/$[Co(tpy)_2]^{2+}$ couple (Supplementary Fig. 16). These potential-dependences of conductivity also indicated the electron-hopping-based charge-transfer mechanism[32,43]. Furthermore, the lower conductivity than **1** at the same potential region[32] suggested that the electron hopping between $[Ni(mnt)_2]^{n-}$ and $[Co(tpy)_2]^{m+}$ sites is critical in the conductivity. These results indicate that the conductivity of **2** stems from electron hopping between redox active metal complex sites in **2**.

Further investigation on the conductivity mechanism of **2** was performed through electrochemical analysis (Supplementary Fig. 17). The cyclic voltammograms of **2** revealed a redox wave corresponding to the $[Co(tpy)_2]^{2+}$/$[Co(tpy)_2]^+$ redox couple at $-1.14$ V vs. $Fc^+/Fc$. Notably, the reduction wave of the redox couple decreased in subsequent cycles, stabilizing after the second cycle. This behaviour was not seen in the cyclic voltammograms of **3**, which is almost identical to the cyclic voltammogram of **1**. The decrease in the redox wave intensity in **2** can be ascribed to the charge-trapping phenomena of the redox couple of $[Ni(mnt)_2]^-$/$[Ni(mnt)_2]^{2-}$ mediated by the electron transfer based on electron hopping mechanism through the cationic framework[32]. If the band-like electron transfer between the $[Ni(mnt)_2]^{n-}$ sites is critical to the electronic conductivity in **2**, the redox wave of the $[Ni(mnt)_2]^-$/$[Ni(mnt)_2]^{2-}$ couple would be directly observed in the cyclic voltammogram. Therefore, the presence of the charge trapping effect also indicates that the electron hopping between $[Co(tpy)_2]^{m+}$ and $[Ni(mnt)_2]^{n-}$ sites based on redox-conduction mechanism is a plausible electron transport pathway in **2** (Fig. 5c).

## Conclusions

In conclusion, anion-exchange in bis(terpyridine)cobalt(II) polymer films leads to significant change in their solid-state electronic behaviour on electrodes, altering capacitor and conductor states. The chloride-containing coordination polymer, **1**, acts as a solid-state electrolyte, facilitating the formation of electrochemical supercapacitors. However, when the monovalent anionic metalladithiolene $[Ni(mnt)_2]^-$ was embedded in the bis(terpyridine)cobalt(II) polymer, the material exhibits semiconducting properties. This change in electronic behaviour is due to the electronic interactions between the two redox-active components with similar redox potentials. This is supported by Raman and XP spectroscopy along with electrochemical analysis. Furthermore, the bis(terpyridine)cobalt(II) polymer containing the divalent anionic metalladithiolene $[Ni(mnt)_2]^{2-}$ exhibits insulating characteristics, owing to the absence of charge transfer interaction. Our findings suggest that bis(terpyridine)metal(II) polymer films represent a promising platform for electronic materials, offering the ability to fine-tune of host-guest electronic interactions via anion-exchange.

## Methods
### Materials

**1** and $(nBu_4N)_2[Ni(mnt)_2]$ were prepared according to the previous literature, respectively[31,40]. $(nBu_4N)[Ni(mnt)_2]$, $(nBu_4N)[Ni(tdt)_2]$, and $nBu_4NCl$ were purchased from Tokyo Chemical Industry Co., Ltd., and used as received. Dehydrated $CH_3CN$ and $C_2H_5OH$ were purchased from Kanto Chemical Co., Inc., and used without further purification.

$nBu_4NClO_4$ and $nBu_4NPF_6$ used for electrochemical analysis were purchased from Tokyo Chemical Industry Co., Ltd. and FUJIFILM Wako Pure Chemical Corporation, respectively, and purified by recrystallization from hot ethanol.

### Apparatus

AFM measurements were performed on Agilent Technologies 5500 Scanning Probe Microscope. AFM was carried out using a silicon cantilever NCH (Nano World) in the high amplitude mode (Tapping Mode) under an ambient condition. SEM/EDS measurements were performed on JEOL JCM7000 electron microscope equipped with EDS analyser, which was operated with an acceleration voltage at 15 kV. Cross-sectional SEM/EDS measurements were performed using JEOL JSM-7800F-PRIME electron microscope with an acceleration voltage of 15 kV. The samples for the SEM/EDS measurements were fabricated with the SM-09020CP and SM-09010CP Cross-section Polisher. XPS was recorded with ALVAC PHI VersaProbe 5000 and VersaProbe III spectrometers with a monochromatic Al Kα X-ray source (15 kV, 25 W). The spectra were standardized using a C 1s peak of the adventitious carbon at 284.6 eV. All I-V curve measurements were recorded with BAS ALS 750E and HOKUTO HZ-Pro S4 electrochemical analysers. Cyclic voltammetry was recorded with the BAS ALS 750E electrochemical analyser.

### Solid-state I-V measurements

The responses to external electric field of **1**–**4** were measured with Au IDA electrodes (Micrux Technologies, ED-IDE2-Au). These electrodes feature a 5 μm gap between two fingers and a 10 μm finger width. For measurements, a flake of **1** was dropcasted on the IDA electrode, and an anion-exchange reaction was then conducted on the electrodes. All the samples were dried under vacuum before the measurements. The areal volumetric capacitances were calculated based on the effective area of the films, which was approximated as one-third of the entire films, considering the gap and finger width mentioned above. Temperature-dependent conductivity measurements were performed in an Ar-filled glove box.

### Electrochemical measurements

Cyclic voltammograms were obtained under an Ar atmosphere using a standard 3 electrode setup. F-doped $SnO_2$ (FTO) electrode modified with **1**, **2** and **3** were used as a working electrode. Pt wire was used as a counter electrode. A homemade $Ag^+$/Ag reference electrode (0.01 M $AgClO_4$ in 0.1 M $nBu_4NClO_4$/$CH_3CN$) was used, and the reported potentials were standardized according to the external $Fc^+/Fc$ redox couple measured under the same experimental conditions.

### Potential-dependent conductivity measurements

Potential-dependent conductivity was measured with Au interdigitated array electrode (BAS Inc., 012125), which consists of two working electrodes. The electrode features a 5 μm gap between two fingers and a 10 μm finger width. For measurements, a flake of **1** was dropcasted on the IDA electrode, and an anion-exchange reaction was then conducted on the electrodes. All the samples were dried under vacuum before the measurements. The electrochemical experiments were performed in 0.1 M $nBu_4NPF_6$/$CH_3CN$ under an Ar atmosphere. The I-V curves between the working electrodes were recorded with different potential. Au and AgCl/Ag ink (BAS Inc., 011464) was used as a counter electrode and reference electrode, respectively, and the reported potentials were standardized according to the external $Fc^+/Fc$ redox couple measured under the same experimental conditions.

### Preparation of 2

**1** on substrates (e.g. Si and carbon paper) were immersed in a saturated $(nBu_4N)[Ni(mnt)_2]$ solution in $C_2H_5OH$ in a glass vial, and the reaction container was kept undisturbed for 3 days. After the reaction, the substrates were washed with $CH_3CN$ and $C_2H_5OH$ successively, and dried under $N_2$ blow.

https://doi.org/10.1038/s42004-024-01274-4 **Article**

## Preparation of 3

Under an Ar atmosphere, **1** on substrates (e.g. Si and carbon paper) were immersed in a 5 mM $(nBu_4N)_2[Ni(mnt)_2]$ solution in $CH_3CN$ in a glass vial, and the reaction container was kept undisturbed for 3 days. After the reaction, the substrates were washed with $CH_3CN$ and $C_2H_5OH$ successively, and dried under Ar.

## Preparation of 4

**1** on substrates (e.g. Si and carbon paper) were immersed in a saturated $(nBu_4N)[Ni(tdt)_2]$ solution in $CH_3CN$ in a glass vial, and the reaction container was kept undisturbed for 3 days. After the reaction, the substrates were washed with $CH_3CN$ and $C_2H_5OH$ successively, and dried under $N_2$ blow.

## Inverse anion-exchange reaction of 2

**2** on Si substrates were immersed in a 5 mM $nBu_4NCl$ solution in $C_2H_5OH$ in a glass vial, and the reaction container was kept undisturbed for 3 days. After the reaction, the substrates were washed with $CH_3CN$ and $C_2H_5OH$ successively, and dried under $N_2$ blow.

## Inverse anion-exchange reaction of 3

Under an Ar atmosphere, **3** on Si substrates were immersed in a 5 mM $nBu_4NCl$ solution in $CH_3CN$ in a glass vial, and the reaction container was kept undisturbed for 3 days. After the reaction, the substrates were washed with $CH_3CN$ and $C_2H_5OH$ successively, and dried under $N_2$ blow.

## Data availability

The data supporting the findings of this study are available within this article and its Supplementary Information. The data related to the figures in the main text and Supplementary Materials are provided as Excel files in Supplementary Data 1 and 2, respectively.

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

## Acknowledgements

This research received financial supports from the Japan Society for the Promotion of Science (JSPS KAKENHI) under Grant Nos.: JP19H05460, JP20K15242, JP22K05055, and JP24H00468, as well as the White Rock Foundation. Additionally, we acknowledge the support of the Advanced Research Infrastructure for Materials and Nanotechnology in Japan (ARIM) of the Ministry of Education, Culture, Sports, Science and Technology (MEXT) (JPMXP09-A-21-UT-0027, JPMXP1222UT0007, JPMXP1223UT0025, and JPMXP1224UT0037) for XPS measurements and cross-sectional SEM/EDS analysis.

## Author contributions

K.T. initiated the research, and performed the synthesis, measurements, and analyses. M.I. and N.F. conducted the temperature-dependent conductivity measurement. H.N. supervised the research project. K.T. and H.N. wrote the manuscript, and all authors reviewed and approved the final version.

## Competing interests

The authors declare no competing interests.
