## [Peer Review File · Communications Chemistry]

Reviewers' comments:

Reviewer #1 (Remarks to the Author):

Hiroshi Nishihara and co-workers synthesized redox active bis(terpyridine)cobalt(II) polymers and discussed their supercapacitor properties and redox-state-dependent conductor behavior. I have detailed major concerns/comments that should be addressed.

1. My chief concern is the lack of characterization for conductivity mechanism of 2 and currently there is not conclusive experimental evidence for the partial charge-transfer model.

2. The description of “electric field responses” should be better discussed.

The modulation between conductive and insulative was achieved by anion-exchange, rather than electric field.

3. how to achieve reversible change between conductive and insulative by the in-situ stimulation of electric field.

Reviewer #2 (Remarks to the Author):

Kenji Takada et al. reported an interesting bis(terpyridine)cobalt(II) system based on ion-exchange, and the conductivity of the polymer varies significantly depending on the anion. This work showed some interesting properties, however, there are some issues that remain to be resolved. Therefore, I would recommend it published in Communications Chemistry after minor revision.

1. EDS and XPS are generally used to analyze elemental information on the surface of materials, so more evidence of the anion content inside the film after ion exchange is required. Is it possible to disassemble the polymer to detect the ionic composition.

2. I would suggest that authors should provide more explanation about the drivers of anion exchange.

3. I would suggest authors should provide more information on the synthesis and characterization data of polymer 1.

4. Three-dimensional porous materials have become the focus of research in supercapacitors in recent years, and authors should cite articles in this area, such as Chem 2017, 2, 171-200.

Reviewer #1

Hiroshi Nishihara and co-workers synthesized redox active bis(terpyridine)cobalt(II) polymers and discussed their supercapacitor properties and redox-state-dependent conductor behavior. I have detailed major concerns/comments that should be addressed.

We greatly appreciate for reviewing our manuscript and giving encouraging comments. We have updated our manuscript according to your comments, in which the revised points are highlighted with blue letters. And point-by-point responses to your concerns are provided below. We hope that the manuscript has been thoroughly improved by your reviewing.

1. My chief concern is the lack of characterization for conductivity mechanism of **2** and currently there is not conclusive experimental evidence for the partial charge-transfer model.

We appreciate the reviewer's comment. To clarify the conductivity mechanism in **2**, we performed detailed *I-V* curve measurements and analyzed the results based on theory of redox conduction (SurrIDGE, N. A. et al. *J. Phys. Chem.* **96**, 962–970 (1992)). As a result, the *I-V* curve between -6 V and +6V was not linear. This non-linear *I-V* curve can be fitted by the simulation of redox conductivity as shown in Fig. A-1. These results indicate that the conductivity mechanism in **2** is redox conduction, dominated by electron hopping between two neighboring redox sites with different oxidation states. In addition, we also performed the potential-dependent conductivity measurements of **2**. As shown in the results (Fig. A-2), **2** showed increasing in electronic conductivity around the redox potentials of $[\text{Co}(\text{tpy})_2]^{2+}$ and $[\text{Ni}(\text{mnt})_2]^-$. The results also suggested the electron hopping-based conductivity mechanism.

In polymer **2**, charge transfer interaction between $[\text{Co}(\text{tpy})_2]^{2+}$ moieties and $[\text{Ni}(\text{mnt})_2]^-$ generates $[\text{Co}(\text{tpy})_2]^{3+}$ moieties and $[\text{Ni}(\text{mnt})_2]^{2-}$, which was experimentally confirmed by XPS and Raman spectroscopy (Fig. 3). Thus, both $[\text{Co}(\text{tpy})_2]^{2+}$ and $[\text{Co}(\text{tpy})_2]^{3+}$, and $[\text{Ni}(\text{mnt})_2]^-$ and $[\text{Ni}(\text{mnt})_2]^{2-}$ states coexist in polymer **2**. That is, neighboring redox sites in the different oxidation states for redox conduction are formed. This is the plausible conduction mechanism of polymer **2**.

We have added the Figs. A-1 and A-2 as Supplementary Figs. 15 and 16, respectively, and revised the explanation for the experimental results and conduction mechanism in the main text (the third paragraph in p. 8) as follows and Supplementary Note 1 in the Supplementary material.

“To investigate the conductivity mechanism of **2**, the *I-V* curve was measured with wider voltage range between -6 V and +6 V (Supplementary Fig. 15a). In the wider range, the current was not linear to the applied voltage (Supplementary Fig. 15b). This non-linear *I-V* curves were fitted with the simulation based on the redox conduction mechanism.⁴⁷ In addition, the potential-dependent conductivity measurement of **2** shows the conductivity increasing around -0.1 V vs. Fc^+/Fc , near to the $[\text{Co}(\text{tpy})_2]^{3+}/[\text{Co}(\text{tpy})_2]^{2+}$ couple (Supplementary Fig. 16). These potential-dependence of conductivity also indicated the electron-hopping-based charge-transfer mechanism.^{32,48} Furthermore, the lower conductivity than **1** at the same potential region³² suggested that the electron hopping between $[\text{Ni}(\text{mnt})_2]^{n-}$ and $[\text{Co}(\text{tpy})_2]^{m+}$ sites is critical in the conductivity. These results indicate that the conductivity of **2** stems from electron hopping between redox active metal complex sites in **2**.”

Fig. A-1. Electrochemical conductivity of **2**. (a) Experimental I - V curve of **2** between -6 V and $+6$ V (solid red line) and simulated I - V curve based on the hopping mechanism (black dotted line). (b) Comparison of I - V plot of **2** in a (red circle) and simulated linear I - V plot (black dotted line).

Fig. A-2. Potential-dependent conductivity of **2**. (0.1 M $n\text{Bu}_4\text{NPF}_6$ in CH_3CN) Conductivity measurements were performed from the cathodic potential regions.

2. The description of “electric field responses” should be better discussed.

The modulation between conductive and insulative was achieved by anion-exchange, rather than electric field.

We appreciate the reviewer’s comment. As you pointed, the modulation between capacitor and conductor of coordination polymers **1**, **2**, and **3** was achieved by anion-exchange reaction. Application of external electric field is not used for the modulation. The “electric-field response” in our manuscript means the behavior of these coordination polymers on interdigitate array electrodes under application of external voltage. The external voltage is not the trigger for capacitor-conductor transition.

For better understanding, we altered “electric-field responses” to “responses to external electric field” in the manuscript.

3. how to achieve reversible change between conductive and insulative by the in-situ stimulation of electric

field.

We are grateful for the reviewer's constructive suggestion. In our system, the conductive and insulative nature was determined by the anions encapsulated in the coordination polymers, not by in situ stimulation of electric field. In this meaning, the modulation of electronic functions is *ex-situ* modulation.

In-situ modulation of electronic properties by electric field is an important and interesting topic. In the case of inorganic thin film materials, it has been reported that anion migration induced by external electric field leads to modulation in electronic and optical properties (Rasouli, H. R. et al. *Nano Lett.* **21**, 3997–4005 (2021). etc.). Therefore, to realize such *in-situ* modulation, encapsulation of redox-active and mobile anions inside the polymer film is a possible strategy. Such *in-situ* tuning is beyond the scope of the current manuscript, which focuses on the *ex-situ* modulation by anion-exchange, and this is future challenge.

For better understanding of our results, we revised the last sentence in the introduction section as follows: "Our findings suggest that anion-exchange reaction with bis(terpyridine)cobalt(II) polymers is a practical method for precise *ex-situ* control of their electronic functions."

Reviewer #2

Kenji Takada et al. reported an interesting bis(terpyridine)cobalt(II) system based on ion-exchange, and the conductivity of the polymer varies significantly depending on the anion. This work showed some interesting properties, however, there are some issues that remain to be resolved. Therefore, I would recommend it published in Communications Chemistry after minor revision.

We deeply appreciate the reviewer's insightful comments and positive recommendation for publication after revision. We have updated our manuscript according to your comments, in which the revised points are highlighted with blue letters. And point-by-point responses to your concerns are provided below. We hope that the manuscript has been thoroughly improved by your reviewing.

1. EDS and XPS are generally used to analyze elemental information on the surface of materials, so more evidence of the anion content inside the film after ion exchange is required. Is it possible to disassemble the polymer to detect the ionic composition.

We appreciate the reviewer's valuable comments. We agree that SEM/EDS and XPS are surface specific analyses. To investigate the efficiency of anion-exchange reaction inside the film, we have performed cross-sectional elemental analysis of **2** and **3** by SEM/EDS (Fig. B-1). These SEM/EDS elemental mappings confirmed that the anion-exchange reaction occurred inside the film, not limited at the surface of **1**. We have added Fig. B-1 as Supplementary Fig. 3 in the Supplementary Materials, and also added the explanation for these results to the manuscript in the second paragraph in p. 3 as follows.

“The cross-sectional elemental mapping by scanning transmission electron microscopy confirmed that Cl was not detected in the polymer films (Supplementary Fig. 3), meaning that the anion-exchange reaction efficiently proceeded without the limitation at the surface of **1**.”

Fig. B-1. Cross-sectional SEM/EDS elemental mapping and spectra of **2** (a) and **3** (b).

2. I would suggest that authors should provide more explanation about the drivers of anion exchange.

We appreciate the reviewer's insightful suggestion. To investigate the anion-exchange reaction, we have performed the inverse anion-exchange reactions from **2** to **1** and **3** to **1**. As the results, both $[\text{Ni}(\text{mnt})_2]^-$ and $[\text{Ni}(\text{mnt})_2]^{2-}$ were partially replaced with Cl^- (Fig. B-2). These results indicate that the incorporation of the

nickelladithiolene anions inside the bis(terpyridine)cobalt(II) polymer is thermodynamically more favorable than chloride, which drives the anion-exchange reactions from **1** to **2** and **1** to **3**. We have added Fig. B2 as Supplementary Fig. 8 in the Supplementary Materials and also added the explanation on these results to the manuscript in the second paragraph in p. 5 as follows.

“To investigate the driving force of the anion-exchange reaction from chloride to $[\text{Ni}(\text{mnt})_2]^{n-}$, the inverse anion-exchange reaction was performed for **2** and **3** with 5 mM $n\text{Bu}_4\text{NCl}$ solution. The SEM/EDS revealed that the coexistence of metalladithiolene anions and chloride after the reaction, with the approximately 60% and 70% anion-exchange, respectively (Supplementary Fig. 8). The excess Cl^- did not replace the metalladithiolenes completely. These results indicated that the anion-exchange from chloride to $[\text{Ni}(\text{mnt})_2]^{n-}$ was a thermodynamically favoured reaction. π - π interactions and charge-transfer interactions between the bis(terpyridine)cobalt(II) polymer backbone and the metalladithiolenes are preferable while chloride- π interactions were less interactive,³⁸ which is the plausible origin of the driving force to the anion-exchange reaction.”

Fig. B-2. SEM/EDS spectra after the inverse anion-exchange with Cl^- for **2** (a) and **3** (b).

3. I would suggest authors should provide more information on the synthesis and characterization data of polymer 1.

We appreciate the reviewer's suggestion to add the synthesis and characterization of **1**. We have added a following paragraph explaining the synthesis and characterization of **1** at the beginning of the "Results and Discussion" section. We have also added the characterization data including SEM/EDS elemental mapping results, Raman spectrum, XP spectra and cyclic voltammogram of **1** as Supplementary Figs. 1 and 2 in the Supplementary Materials.

4. Three-dimensional porous materials have become the focus of research in supercapacitors in recent years, and authors should cite articles in this area, such as Chem 2017, 2, 171-200.

We appreciate the reviewer's suggestion. We have added two reference articles related to the supercapacitors with porous materials such as metal-organic frameworks and porous nanocarbon materials (Xu, X. et al. *ACS Appl. Mater. Interfaces* **9**, 38737–38744 (2017) and Yao, X. et al. *Chem* **2**, 171–200 (2017)) as references 41 and 42, respectively.

REVIEWERS' COMMENTS:

Reviewer #1 (Remarks to the Author):

accepted

Reviewer #2 (Remarks to the Author):

The author's experiments, as well as the explanations, have completely cleared up my doubts, and I recommend that the manuscript be published as soon as possible.

Reviewer #1

Accepted.

We greatly appreciate for reviewing our manuscript and recommendation for publications in *Communications Chemistry*.

Reviewer #2

The author's experiments, as well as the explanations, have completely cleared up my doubts, and I recommend that the manuscript be published as soon as possible.

We greatly appreciate for reviewing our manuscript and strong recommendation for publications in *Communications Chemistry*.